# Risk factors for gastric perforation after cytoreductive surgery in patients with peritoneal carcinomatosis: Splenectomy and increased body mass index

**Martina Aida Angeles**[1☯‡], **Carlos Martínez-Gómez**[1,2☯‡], **Mathilde Del**[1], **Federico Migliorelli**[3], **Manon Daix**[1], **Anaïs Provendier**[1], **Muriel Picard**[4], **Jean Ruiz**[4], **Elodie Chantalat**[1], **Hélène Leray**[1], **Alejandra Martinez**[1,2], **Laurence Gladieff**[5], **Gwénaël Ferron**[1,6]*

1 Department of Surgical Oncology, Institut Claudius Regaud – Institut Universitaire du Cancer de Toulouse – Oncopole, Toulouse, France, 2 INSERM CRCT Team 1, Tumor Immunology and Immunotherapy, Toulouse, France, 3 Department of Gynecology and Obstetrics, Centre Hospitalier Intercommunal des Vallées de l'Ariège, St Jean de Verges, France, 4 Intensive Care Unit, Institut Claudius Regaud – Institut Universitaire du Cancer de Toulouse – Oncopole, Toulouse, France, 5 Department of Medical Oncology, Institut Claudius Regaud – Institut Universitaire du Cancer de Toulouse – Oncopole, Toulouse, France, 6 INSERM CRCT Team 19, ONCOSARC – Oncogenesis of sarcomas, Toulouse, France

☯ These authors contributed equally to this work.
‡ These authors share first authorship on this work.
* Ferron.Gwenael@iuct-oncopole.fr

## Abstract

### Background

Gastric perforation after cytoreductive surgery (CRS) is an infrequent complication. There is lack of evidence regarding the risk factors for this postoperative complication. The aim of this study was to assess the prevalence of postoperative gastric perforation in patients undergoing CRS for peritoneal carcinomatosis (PC) and to evaluate risk factors predisposing to this complication.

### Methods

We designed a unicentric retrospective study to identify all patients who underwent an open upfront or interval CRS after a primary diagnosis of PC of different origins between March 2007 and December 2018 at a French Comprehensive Cancer Center. The main outcome was the occurrence of postoperative gastric perforation.

### Results

Five hundred thirty-three patients underwent a CRS for PC during the study period and 13 (2.4%) presented a postoperative gastric perforation with a mortality rate of 23% (3/13). Neoadjuvant chemotherapy was administered in 283 (53.1%) patients and 99 (18.6%) received hyperthermic intraperitoneal chemotherapy (HIPEC). In the univariate analysis, body mass index (BMI), peritoneal cancer index, splenectomy, distal pancreatectomy, and

**Data Availability Statement:** All relevant data are within the manuscript and its Supporting information files.

**Funding:** The author(s) received no specific funding for this work.

**Competing interests:** The authors have declared that no competing interests exist.

**Abbreviations:** BMI, body mass index; CC-score, completeness cytoreduction score; CI, confidence interval; CRS, cytoreductive surgery; CT, computed tomography; DMPM, diffuse malignant peritoneal mesothelioma; HIPEC, hyperthermic intraperitoneal chemotherapy; ICU, Intensive Care Unit; NACT, neoadjuvant chemotherapy; OR, odds ratio; PC, peritoneal carcinomatosis; PCI, peritoneal cancer index; PMP, pseudomyxoma peritonei.

histology were significantly associated with postoperative gastric perforation. After multivariate analysis, BMI (OR [95%CI] = 1.13 [1.05–1.22], p = 0.002) and splenectomy (OR [95% CI] = 26.65 [1.39–509.67], p = 0.029) remained significantly related to the primary outcome.

## Conclusions

Gastric perforation after CRS is a rare event with a high rate of mortality. While splenectomy and increased BMI are risk factors associated with this complication, HIPEC does not seem to be related. Gastric perforation is probably an ischemic complication due to a multifactorial process. Preventive measures such as preservation of the gastroepiploic arcade and prophylactic suture of the greater gastric curvature require further assessment.

## Introduction

Peritoneal carcinomatosis (PC) is the dissemination within the abdominal cavity of any form of cancer, whether or not it originated from the peritoneum itself, and is most commonly caused by abdominopelvic malignancies [1]. Depending on the origin of the malignancy, cytoreductive surgery (CRS) represents the standard of care in order to remove all macroscopic disease [2–5], including different surgical procedures such as extended peritonectomy, infragastric omentectomy, splenectomy, distal pancreatectomy, atypical partial gastrectomy, cholecystectomy, and Hudson procedure [6–8]. In some malignancies, hyperthermic intraperitoneal chemotherapy (HIPEC) is associated with CRS, as a survival benefit has been described [3, 5, 9, 10]. Since the recent publication of a randomized trial in stage III ovarian cancer, which showed that the addition of HIPEC to CRS provided a higher recurrence-free and overall survival rate after three cycles of NACT, HIPEC has been introduced to clinical practice [9]. HIPEC is also the gold standard for pseudomyxoma peritonei (PMP) and diffuse malignant peritoneal mesothelioma (DMPM) [11, 12].

Cytoreductive procedures have been associated with non-negligible postoperative morbimortality rates, with around 20% of grade III/IV surgical complications [13] and a postoperative mortality rate of approximately 3% [14]. Morbimortality seems to increase with the association of CRS plus HIPEC [15, 16]. Different types of postoperative complications such as pleural effusion, pneumonia, intra-abdominal collection or abscess, bleeding, bowel anastomotic leakage, bowel perforation, and pancreatic fistula have been described. However, there are very few reports in the literature assessing the prevalence of gastric perforations after CRS. While some authors state that its occurrence is strongly associated with HIPEC, evidence regarding other associated risk factors is missing [17, 18].

The aim of our study was to assess the prevalence of postoperative gastric perforation in patients undergoing a CRS for PC of different origins and to evaluate the risk factors predisposing to this complication.

## Materials and methods

### Patient selection and study design

A computer-generated search in the institutional patient database was carried out in February 2020 to retrospectively identify all patients who underwent an open upfront or interval CRS after primary diagnosis of PC of different origins (ovarian cancer, endometrial cancer, colon cancer, PMP and DMPM) between March 2007 and December 2018 at the Institut Claudius

Regaud Comprehensive Cancer Center—IUCT—Oncopole (Toulouse, France), which is an expert center for rare peritoneal diseases (RENAPE network). Patients undergoing a secondary CRS for recurrence of PC were excluded from our study. As well, patients with a previous incomplete surgery performed outside our institution and undergoing CRS at our center were excluded. All data that could possibly be used to identify individual patients was deleted and all records were anonymized during the retrieval procedure, before the final database was handed to the researchers. Institutional Review Board (*Comité d'Ethique de Recherche Clinique*) approval was obtained from our center.

## Surgical technique

All surgical procedures were performed by three experienced oncological surgeons, using an open approach with a midline xyphopubic incision. The extent and spread of the disease throughout the 13 abdominopelvic regions were evaluated using the peritoneal cancer index (PCI) [19] and the cytoreductive surgical technique was performed following Surgarbaker's principles of peritonectomy [6]. In case of remnant millimetric lesions in the mesentery or bowel serosa, visceral peritoneal destruction was performed using an electrosurgical ball-tip [20]. The main goal of the surgery was to obtain complete cytoreduction, evaluated using the Completeness of Cytoreduction score (CC-0: No residual tumor; CC-1: Residual disease less than 2.5 mm in diameter; CC-2: Residual nodules between 2.5 mm and 2.5 cm; and CC-3: Residual nodules greater than 2.5 cm or a confluence of unresectable disease) [19]. HIPEC was performed after CRS using the open coliseum technique with different drugs and protocols depending on the pathology. Infragastric omentectomy without preservation of the gastroepiploic arcade was performed using non-absorbable polymer locking clips (Hem-O-Lok®, Weck Closure Systems, Research Triangle Park, NC). In more recent years, in order to decrease the incidence of postoperative gastric perforation after CRS in patients undergoing an infragastric omentectomy combined with a splenectomy, two additional surgical techniques have been implemented in selected cases at the surgeon's discretion. First, we try to preserve the gastroepiploic arcade when performing an infragastric omentectomy, if disease is absent at this localization (S1 Fig). Second, a prophylactic suture of the greater curvature of the stomach is performed, consisting in a seromuscular plication that may prevent gastric perforation, as it has been suggested by other authors (S2 Fig) [17]. Finally, proton-pump inhibitors were systematically administered in the postoperative period.

## Study data

The main outcome was the occurrence of gastric perforation in the postoperative period. Patient demographic data (age, gender, body mass index [BMI], diabetes mellitus), neoadjuvant chemotherapy (NACT), PCI scores calculated during surgery, selected procedures performed during CRS that could have been related to postoperative gastric perforation (infragastric omentectomy with or without preservation of the gastroepiploic arcade, splenectomy, distal pancreatectomy, atypical partial gastrectomy, celiac lymph node resection, prophylactic suture of the greater gastric curvature), HIPEC and histological type were included in the database.

Extended and comprehensive data collection was performed in the patients presenting with postoperative gastric perforation in order to obtain a detailed description of each case.

## Statistical analysis

Data were summarized by frequencies and percentages for categorical variables and by medians and ranges for continuous variables. Univariate analysis was performed using Fisher's

exact test and Wilcoxon's rank-sum test for categorical and continuous variables, respectively. The characteristics that showed a significant association with the prevalence of gastric perforation during the previous analysis were included in a multivariate logistic regression model, from which odds ratios (OR) and their 95% confidence intervals (CI) were calculated. $p$-values below 0.05 were considered statistically significant. All statistical analyses were conducted using STATA 13.0 software.

## Results

Five hundred thirty-three patients were included in our study. Among them, 13 (2.4%) patients experienced postoperative gastric perforation. The overall median age of the patients was 61.7 years (range 22.0–84.2) and the median BMI was 23.5 kg/m$^2$ (range 14.3–53.4). There were 513 women in the cohort (96.3%) and 32 (6.0%) patients had medical history of diabetes mellitus.

All patients underwent a CRS, 429 (80.5%) for ovarian cancer, 25 (4.7%) for endometrial cancer, 12 (2.3%) for colon cancer, 40 (7.5%) for PMP, and 27 (5.1%) for DMPM. Two hundred eighty-three (53.1%) patients were treated with NACT before CRS. The median PCI in the cohort was 13 (range: 0–39) and 99 (18.6%) patients received HIPEC at the end of CRS. Table 1 summarizes the surgical procedures performed during CRS. All patients were considered CC-0 or CC-1 at the end of CRS.

In univariate analysis, BMI, PCI, splenectomy, distal pancreatectomy, and histology were significantly associated with postoperative gastric perforation occurrence (Table 2). However, after multivariate analysis, only BMI and splenectomy remained significantly related to the primary outcome (Table 3).

Among the thirteen patients with a gastric postoperative perforation, the median age was 65.4 years (range 33.9–80.2) and the median BMI was 27.1 kg/m$^2$ (range 20.2–53.3). Six patients were diagnosed with high grade serous ovarian carcinoma, 5 patients had PMP, 1 patient had DMPM, and 1 patient presented with endometrial clear cell carcinoma. NACT was administered in 5 patients before CRS and the median PCI was 24 (range 13–35). All patients underwent an infragastric omentectomy combined with a splenectomy, and the gastroepiploic arcade was not preserved in any of them. A prophylactic suture of the greater curvature of the stomach was performed in one patient. HIPEC was performed in 5 patients using a protocol based on oxaliplatin 360mg/m$^2$ during 30 minutes using the coliseum technique. At the end of the surgery a nasogastric tube without suction was placed in all patients. The median operative time was 323 minutes (range 200–602). The median time to diagnosis of the perforation was 5 days (range 2–15). The clinical presentation of our patients was a combination of the following signs and symptoms: Acute and severe abdominal pain, abdominal tenderness, nausea, vomiting, gastric fluid in the abdominal drain, fever and/or clinical deterioration. In

**Table 1. Surgical data of all patients included in the study (n = 533).**

| Surgical procedures, n (%) | |
|---|---|
| Infragastric omentectomy | 533 (100) |
| Splenectomy | 192 (36.0) |
| Distal pancreatectomy | 60 (11.3) |
| Celiac lymph node resection | 86 (16.1) |
| Partial gastrectomy | 10 (1.9) |
| Preservation of the gastroepiploic arcade | 28 (5.3) |
| Prophylactic suture of the greater curvature of the stomach | 13 (2.4) |

**Table 2. Factors associated with gastric perforation: Univariate analysis.**

| Patient characteristics | Postoperative gastric perforation | | p-value |
|---|---|---|---|
| | Yes (n = 13) | No (n = 520) | |
| **Age** (years), *median (range)* | 65.4 (33.2–80.2) | 61.6 (22.0–84.2) | *0.110* |
| **Female gender**, *n (%)* | 12 (92.3) | 501 (96.4) | *0.395* |
| **Diabetes mellitus**, *n (%)* | 2 (15.4) | 30 (5.8) | *0.180* |
| **Body Mass Index** (kg/m$^2$), *median (range)* | 27.1 (20.2–53.3) | 23.5 (14.3–48.4) | ***0.014*** |
| Missing | *0* | *5* | |
| **Neoadjuvant chemotherapy**, *n (%)* | 5 (38.5) | 278 (53.5) | *0.400* |
| **PCI**, *median (range)* | 24 (13–35) | 13 (0–39) | *<**0.001*** |
| Missing | *0* | *123* | |
| **Surgical procedures**, *n (%)* | | | |
| Infragastric omentectomy | 13 (100) | 520 (100) | - |
| Splenectomy | 13 (100) | 179 (34.4) | *<**0.001*** |
| Distal pancreatectomy | 5 (38.5) | 55 (10.6) | ***0.010*** |
| Celiac lymph node resection | 2 (15.4) | 84 (16.2) | *1.000* |
| Partial gastrectomy | 0 (0) | 10 (1.9) | *1.000* |
| Preservation of the gastroepiploic arcade | 0 (0) | 28 (5.4) | *1.000* |
| Prophylactic suture of the greater gastric curvature | 1 (7.7) | 12 (2.3) | *0.277* |
| **HIPEC**, *n (%)* | 5 (38.5) | 94 (18.1) | *0.074* |
| **Histology** | | | ***0.005*** |
| Ovarian cancer | 6 (46.2) | 423 (81.4) | |
| Endometrial cancer | 1 (7.7) | 24 (4.6) | |
| Colon cancer | 0 (0) | 12 (2.3) | |
| PMP | 5 (38.5) | 35 (6.7) | |
| DMPM | 1 (7.7) | 26 (5.0) | |

PCI: Peritoneal cancer index.

HIPEC: Hyperthermic intraperitoneal chemotherapy.

PMP: Pseudomyxoma peritonei.

DMPM: Diffuse malignant peritoneal mesothelioma.

all cases the diagnosis was made using an abdominal computed tomography (CT) and confirmed during surgery. The median perforation size was 10 mm (range 2–30) and in all cases the perforation was located at the upper portion of the greater curvature of the stomach (S3 Fig).

**Table 3. Factors associated with gastric perforation: Multivariate analysis.**

| Variable | OR (CI 95%) | p-value |
|---|---|---|
| Body mass index (kg/m$^2$) | 1.13 (1.05–1.22) | **0.002** |
| PCI | 1.05 (0.98–1.12) | 0.206 |
| Splenectomy | 26.65 (1.39–509.67) | **0.029** |
| Distal pancreatectomy | 1.43 (0.42–4.95) | 0.566 |
| Ovarian histology | 0.47 (0.15–1.50) | 0.205 |

OR: Odds ratio.

CI: Confidence interval.

PCI: Peritoneal cancer index.

All patients received prompt surgical management of the gastric perforation, which consisted in an atypical gastrectomy using an automatic stapler reinforced with a manual gastric suture. Three patients experienced a concomitant gastro-pleural fistula, among whom two required pleural decortication by thoracotomy and a long-term insertion of a dual lumen nasogastric tube. Three (23.1%) patients died at the Intensive Care Unit (ICU) at the 7th, 93rd and 111th postoperative day. The first patient experienced refractory septic shock due to a digestive peritonitis caused by the gastric perforation, which was followed by multivisceral failure. The second patient developed multiple complex fistulas during the postoperative course of gastric perforation surgery, leading to a chronic septic status with secondary multiorgan failure (respiratory and acute kidney injury with prolonged mechanical ventilation and continuous veno-venous hemodiafiltration). Finally, an ethical therapeutic limitation was decided upon in the multidisciplinary meeting and the patient died 93 days after CRS. The third patient also developed multiple digestive fistulas leading to a chronic septic status. Acute massive abdominal bleeding occurred 111 days after CRS and, after a multidisciplinary ethical decision, urgent surgery was not performed, therefore, end of life care was given. The remaining 10 patients were discharged from the ICU after a median hospitalization length of 34 days (range 14–68). The median overall hospitalization length of these 10 patients was 53 days (range 21–98). Table 4 shows a detailed description of patient's characteristics, surgical and follow-up data.

## Discussion

Gastric perforation after CRS is a very rare postoperative complication, with a prevalence of 2.4% in our experience. Our findings show a slightly higher prevalence compared to previously published incidences ranging from 0.3 to 1.9% [17, 18, 21–24]. Its occurrence is associated with a high mortality rate (23%), as three patients died in the ICU following the diagnosis of the complication. The mortality rate in our study is in line with previous reports describing the outcome of patients undergoing surgery for perforated peptic ulcers (deceased in around 20–30% of the cases) [25]. However, most studies focusing on gastric perforation after CRS described that this type of complication was not related to a fatal outcome [17, 18, 21–24], while only one study reported a single death due to sepsis caused by the perforation [26]. Nevertheless, most of these series described isolated cases of postoperative gastric perforations [21–24, 26], whereas only two of them included 4 and 6 events [17, 18]. Therefore, it is highly probable that this postoperative complication, and its related mortality, may be underdiagnosed or underreported.

We found that splenectomy was associated with postoperative gastric perforation, and was performed in all patients who experienced this complication. Similarly, the four cases of gastric perforation after CRS and HIPEC reported by Zappa et al. underwent a greater and lesser omentectomy and a splenectomy without gastric resection [17]. In Kyang et al. study, five out of the six patients with this complication underwent a splenectomy during CRS [18]. Our hypothesis is that gastric perforation could be explained by a reduced blood perfusion of the greater curvature of the stomach due to the ligation of short gastric and gastroepiploic vessels during the splenectomy [27]. In all of our cases, the perforation was found in the upper part of the gastric greater curvature, which corresponds to the abovementioned area of devascularization. Concordantly, previous studies systematically found the perforation to be located at or near the greater curvature, close to the area of the left gastroepiploic vessels [17, 18]. In case of ligation of the short gastric and gastroepiploic vessels, the only remaining vascularization of the stomach is the one provided by the left and right gastric arteries, running along the lesser curvature. Therefore, the upper third of the greater curvature becomes the less vascularized

**Table 4. Description of the 13 patients with postoperative gastric perforation after cytoreductive surgery.**

| Patient number; age; gender. | Diag-nosis | WHO perfor-mance status; BMI; Diabetes mellitus | NACT | PCI | Surgical procedures | HIPEC | Operative time (minutes); CC-score; naso-gastric tube | Interval to diagnosis (days) | Perfo-ration size (mm); conco-mitant gastro-pleural fistula | Hospita-lization length* (days) | Current status |
|---|---|---|---|---|---|---|---|---|---|---|---|
| 1; 33; F | PMP | 0; 35.6; no | No | 13 | Extended peritonectomy, infragastric and supragastric omentectomy, splenectomy, cholecystectomy, mesenteric and bowel vaporization, and hysterectomy. | Yes, oxaliplatin | 294; CC-0 | 3 | 3; yes | 67 | NED 3047 days after surgery |
| 2; 60; F | HGSOC | 0; 53.3; no | Yes | 26 | Extended peritonectomy, infragastric and supragastric omentectomy, splenectomy, rectosigmoid resection, mesenteric and bowel vaporization, pelvic and paraaortic lymphadenectomy, bilateral adnexectomy, and hysterectomy. | No | 412; CC-1 | 5 | 20; yes | 98 | DOD 688 days after CRS |
| 3; 74; F | PMP | 1; 44.9; yes | No | 21 | Extended peritonectomy, infragastric and supragastric omentectomy, splenectomy, distal pancreatectomy, cholecystectomy, mesenteric and bowel vaporization, bilateral adnexectomy, and hysterectomy. | No | 323; CC-1 | 2 | 3; no | 93 | Dead from PCs 93 days after CRS |
| 4; 75; F | HGSOC | 0; 24.2; no | Yes | 25 | Extended peritonectomy, infragastric omentectomy, splenectomy, distal pancreatectomy, rectosigmoid resection, mesenteric and bowel vaporization, pelvic and paraaortic lymphadenectomy, bilateral adnexectomy, and hysterectomy. | No | 355; CC-0 | 4 | 20 and 20; no | 73 | DOD 197 days after CRS |
| 5; 64; F | DMPM | 0; 21.9; no | No | 30 | Extended peritonectomy, infragastric and supragastric omentectomy, splenectomy, distal pancreatectomy, cholecystectomy, mesenteric and bowel vaporization, bilateral adnexectomy and hysterectomy. | Yes, oxaliplatin | 602; CC-0 | 15 | 20; no | 111 | Dead from PCs 111 days after CRS |
| 6; 80; F | HGSOC | 0; 22.3; no | No | 18 | Extended peritonectomy, infragastric and supragastric omentectomy, splenectomy, distal pancreatectomy, rectosigmoid resection, bilateral adnexectomy, and hysterectomy. | No | 200; CC-0 | 7 | 2; no | 95 | DOD 544 days after CRS |

*(Continued)*

**Table 4.** (Continued)

| Patient number; age; gender. | Diag-nosis | WHO perfor-mance status; BMI; Diabetes mellitus | NACT | PCI | Surgical procedures | HIPEC | Operative time (minutes); CC-score; naso-gastric tube | Interval to diagnosis (days) | Perfo-ration size (mm); conco-mitant gastro-pleural fistula | Hospita-lization length* (days) | Current status |
|---|---|---|---|---|---|---|---|---|---|---|---|
| 7; 48; M | PMP | 0; 26.7; no | Yes | 35 | Extended peritonectomy, infragastric and supragastric omentectomy, splenectomy, distal pancreatectomy, ileo-cecal resection, mesenteric and bowel vaporization, celiac lymhadenectomy. Prophylactic suture of the gastric greater curvature. | Yes, oxaliplatin | 550; CC-0 | 2 | 10; no | 37 | DOD 406 days after CRS |
| 8; 63; F | HGSOC | 0; 24.9; no | No | 20 | Extended peritonectomy, infragastric omentectomy, splenectomy, cholecystectomy, rectosigmoid resection, mesenteric and bowel vaporization, pelvic, paraaortic and celiac lymphadenectomy, bilateral adnexectomy and hysterectomy. | No | 232; CC-0 | 2 | 30; no | 48 | DOD 1968 days after CRS |
| 9; 73; F | PMP | 0; 23.8; no | No | 34 | Extended peritonectomy, infragastric omentectomy, splenectomy, cholecystectomy, ileocolic resection, mesenteric and bowel vaporization, pelvic and paraaortic lymphadenectomy, bilateral adnexectomy and hysterectomy | Yes, oxaliplatin | 321; CC-0 | 4 | 20; no | 21 | NED 1520 days after surgery |
| 10; 65; F | HGSOC | 0; 27.8; no | Yes | 16 | Extended peritonectomy, infragastric omentectomy, splenectomy, pelvic and paraaortic lymphadenectomy, bilateral adnexectomy and hysterectomy. | No | 221; CC-0 | 5 | 5; no | 53 | DOD 954 days after CRS |
| 11; 71; F | HGSOC | 0; 31.9; no | Yes | 24 | Extended peritonectomy, infragastric and supragastric omentectomy, splenectomy, distal pancreatectomy, cholecystectomy, transvers colic resection, mesenteric and bowel vaporization, pelvic and paraaortic lymphadenectomy, bilateral adnexectomy, and hysterectomy. | No | 287; CC-0 | 6 | 10; no | 7 | Dead from PCs 7 days after CRS |

(*Continued*)

**Table 4.** (Continued)

| Patient number; age; gender. | Diagnosis | WHO performance status; BMI; Diabetes mellitus | NACT | PCI | Surgical procedures | HIPEC | Operative time (minutes); CC-score; naso-gastric tube | Interval to diagnosis (days) | Perforation size (mm); concomitant gastro-pleural fistula | Hospitalization length* (days) | Current status |
|---|---|---|---|---|---|---|---|---|---|---|---|
| 12; 72; F | ECC | 0; 32.3; yes | No | 18 | Extended peritonectomy, infragastric and supragastric omentectomy, splenectomy, pelvic and paraaortic lymphadenectomy, bilateral adnexectomy, and hysterectomy. | No | 324; CC-0 | 5 | 10; yes | 25 | AWD, recurrence 409 days after CRS |
| 13; 62; F | PMP | 0; 20.2; no | No | 29 | Extended peritonectomy, infragastric and supragastric omentectomy, splenectomy, cholecystectomy, mesenteric and bowel vaporization, pelvic and paraaortic lymphadenectomy, bilateral adnexectomy, and hysterectomy. | Yes, oxaliplatin | 372; CC-0 | 12 | 3; no | 23 | NED 723 days after surgery |

WHO: World Health Organization.

BMI: Body mass index.

NACT: Neoadjuvant chemotherapy.

PCI: Peritoneal carcinomatosis index.

HIPEC: Hyperthermic intraperitoneal chemotherapy.

CC-score: Completeness of cytoreduction score.

* including hospitalization at intensive care unit.

F: Female.

M: Male.

PMP: Pseudomyxoma peritonei.

HGSOC: High-grade serous ovarian carcinoma.

DMPM: Diffuse malignant peritoneal mesothelioma.

ECCC: Endometrial clear cell carcinoma.

PCs: Postoperative complications.

DOD: Dead of disease.

NED No evidence of disease.

AWD: Alive with disease.

CRS: Cytoreductive surgery.

area of the stomach [27]. In this study, we could not analyze if infragastric omentectomy was a risk factor for gastric perforation as all patients underwent this procedure during CRS. Still, associated omentectomy and splenectomy may have an additive effect on the devascularization of the greater curvature of the stomach, as right and left gastroepiploic and short gastric vessels are usually sectioned when performing these procedures together. We did not find distal pancreatectomy to be associated with postoperative gastric perforation, which may be explained by the low number of patients in our series undergoing this procedure. Distal pancreatectomy may increase the devascularization of the greater curvature, thereby favoring the risk of gastric perforation.

According to our results, gastroepiploic arcade preservation and prophylactic suture of the greater gastric curvature over the ligated vessels have not been shown to prevent gastric perforation. However, these procedures have only recently been introduced in this type of surgeries in our center, therefore, only few patients benefited from these prophylactic measures (6% gastroepiploic arcade preservation and 2% prophylactic suture). The rationale for preserving the gastroepiploic arcade is to reduce the devascularization of the greater curvature, thereby decreasing the ischemia of the region and the subsequent risk of postoperative gastric perforation [17]. As previously mentioned, gastric perforations are commonly located in the upper portion of the greater curvature. Therefore, preserving the gastroepiploic arcade during an omentectomy would reduce the risk of perforation, particularly in patients undergoing this procedure combined with a splenectomy. On the other hand, the rationale for a prophylactic suture would be to reinforce the gastric wall and to invaginate the areas of the stomach at higher risk of perforation. In our series, one of the perforated patients underwent a prophylactic suture. However, she had a hiatal hernia and the prophylactic suture performed during the CRS was done in the middle part of the greater curvature instead of in the upper segment, probably being insufficient to prevent the perforation.

Our data have also shown that gastric perforation was associated with high BMI. In fact, for each additional BMI point, the risk of perforation was increased by approximately 13%. This association is biologically plausible as there is large evidence showing that obesity is correlated to deficient wound healing and to other postoperative complications, probably due to inherent anatomic features of adipose tissue, vascular insufficiencies, cellular and composition modifications, oxidative stress, alterations in immune mediators, and nutritional deficiencies [28]. To our knowledge, no previous studies have reported an association between BMI and gastric perforation in patients undergoing CRS [17, 18, 21–24, 26].

Regarding the role of HIPEC, most authors concluded that it was associated with gastric perforation after CRS [17, 18, 21–24, 26]. It has been suggested that HIPEC could have a direct thermal and toxic effect damaging the stomach wall, as well as a systemic effect which would include a temporarily retarded repair of gastric mucosa due to the systemic absorption of the drugs [17]. However, these studies only included patients undergoing CRS plus HIPEC [17, 18, 21–24, 26], without assessing the risk of gastric perforation in patients undergoing CRS alone, of which there are no previous reports. Thus, this postoperative complication may be also present in patients undergoing CRS alone, occurring independently of the addition of HIPEC. Although almost 20% of our cohort underwent HIPEC, we did not find any correlation between the occurrence of postoperative gastric perforation and HIPEC. Along the same lines, we did not find NACT to be associated with gastric perforations.

All patients that experienced gastric perforation in our study were surgically managed with partial gastrectomy in order to remove the ischemic tissue. Even though conservative management of perforated gastric ulcers has been demonstrated to be feasible [29], almost all cases of gastric perforation after CRS reported in the literature have been surgically treated [17, 21, 22, 24]. Moreover, patients with a perforated gastric ulcer still have the omentum which can cover the gastric defect in case of conservative management, whereas patients undergoing CRS have a devascularized greater gastric curvature due to the omentectomy. The high postoperative mortality rate of gastric perforation highlights the importance of promptly assessing any suspicion to avoid the delay in diagnosis. This complication can be clinically suspected if signs of peritonitis or gastric fluid content in intraperitoneal drain are present, or using imaging techniques, such as abdominal CT (S4 Fig). Once diagnosed, gastric perforation should be managed rapidly in order to decrease postoperative mortality rates.

The main strength of our study is the high number of patients included in the analysis. To our knowledge, we report the largest monocentric series of gastric perforation after CRS.

Unlike the other reports, the majority of which only presented data on patients with gastric perforation, we included information about those individuals who did not have this complication, allowing us to statistically analyze the possible risk factors of postoperative gastric perforation, going further than the mere suggestion of possible associations. Moreover, the majority of previous series exclusively included patients who underwent CRS plus HIPEC, whereas we also included patients undergoing CRS alone. Conversely, our study may also have some weaknesses. Its retrospective design may hinder the interpretation of our results as it may introduce biases linked to this type of study. Perioperative nutritional status may be a determining factor for gastric perforation and, due to the retrospective nature of our study, this information could not be assessed. Moreover, we only evaluated the variables that we believed could be related to postoperative gastric perforation, so there might be other factors associated with the outcome which we may have overlooked and unconsciously excluded from the study. Additionally, other postoperative complications than gastric perforations were not assessed in this study.

## Conclusions

In conclusion, gastric perforation after CRS is an infrequent postoperative complication. Splenectomy and high BMI are found to be associated risk factors. However, it is probably a multifactorial process in which many causes may still be unknown. The roles of preservation of the gastroepiploic arcade during infragastric omentectomy -in case of no involvement- and of the prophylactic suture of the greater gastric curvature should be evaluated in further studies.

## Supporting information

**S1 Fig. Preservation of the gastroepiploic arcade during an omentectomy.**
(DOCX)

**S2 Fig. Prophylactic suture of the greater curvature of the stomach.**
(DOCX)

**S3 Fig. Postoperative gastric perforation located in the upper portion of the greater curvature.**
(DOCX)

**S4 Fig. Gastric opacified computed tomography.** Gastric wall perforation and extraluminal free air are marked with yellow arrows.
(DOCX)

## Author Contributions

**Conceptualization:** Martina Aida Angeles, Carlos Martínez-Gómez, Mathilde Del, Federico Migliorelli, Muriel Picard, Jean Ruiz, Elodie Chantalat, Hélène Leray, Alejandra Martinez, Laurence Gladieff, Gwénaël Ferron.

**Data curation:** Martina Aida Angeles, Carlos Martínez-Gómez, Mathilde Del, Federico Migliorelli, Manon Daix, Anaïs Provendier.

**Formal analysis:** Federico Migliorelli.

**Investigation:** Martina Aida Angeles, Carlos Martínez-Gómez, Mathilde Del, Federico Migliorelli, Manon Daix, Anaïs Provendier, Muriel Picard, Jean Ruiz, Elodie Chantalat, Hélène Leray, Alejandra Martinez, Laurence Gladieff, Gwénaël Ferron.

**Methodology:** Martina Aida Angeles, Carlos Martínez-Gómez, Mathilde Del, Federico Migliorelli, Manon Daix, Anaïs Provendier, Muriel Picard, Jean Ruiz, Elodie Chantalat, Hélène Leray, Alejandra Martinez, Laurence Gladieff, Gwénaël Ferron.

**Project administration:** Martina Aida Angeles, Carlos Martínez-Gómez, Federico Migliorelli, Anaïs Provendier, Muriel Picard, Elodie Chantalat, Hélène Leray, Alejandra Martinez, Laurence Gladieff, Gwénaël Ferron.

**Resources:** Martina Aida Angeles, Manon Daix, Gwénaël Ferron.

**Supervision:** Federico Migliorelli, Manon Daix, Anaïs Provendier, Jean Ruiz, Elodie Chantalat, Hélène Leray, Alejandra Martinez, Laurence Gladieff, Gwénaël Ferron.

**Validation:** Martina Aida Angeles, Carlos Martínez-Gómez, Mathilde Del, Federico Migliorelli, Manon Daix, Anaïs Provendier, Muriel Picard, Jean Ruiz, Elodie Chantalat, Hélène Leray, Alejandra Martinez, Laurence Gladieff, Gwénaël Ferron.

**Visualization:** Martina Aida Angeles, Carlos Martínez-Gómez, Mathilde Del, Federico Migliorelli, Manon Daix, Muriel Picard, Jean Ruiz, Elodie Chantalat, Hélène Leray, Alejandra Martinez, Laurence Gladieff, Gwénaël Ferron.

**Writing – original draft:** Martina Aida Angeles, Carlos Martínez-Gómez, Federico Migliorelli.

**Writing – review & editing:** Mathilde Del, Manon Daix, Anaïs Provendier, Muriel Picard, Jean Ruiz, Elodie Chantalat, Hélène Leray, Alejandra Martinez, Laurence Gladieff, Gwénaël Ferron.

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
