## [Decision Letter · Decision Letter 0]

11 Jan 2021

PONE-D-20-33398

Risk factors for gastric perforation after cytoreductive surgery in patients with peritoneal carcinomatosis: splenectomy and increased body mass index

PLOS ONE

Dear Dr. Ferron,

Thank you for submitting your manuscript to PLOS ONE. After careful consideration, we feel that it has merit but does not fully meet PLOS ONE’s publication criteria as it currently stands. Therefore, we invite you to submit a revised version of the manuscript that addresses the points raised during the review process.

The authors present from their CRS database of PC the rare complication of gastric perforation with a UVA that suggests a role for BMI, PCI, splenectomy-distal pancreatectomy and primary histology but with BMI and splenectomy remaining of significance on MVA.

I have several caveats:

1. I would expand a little on the outcome advantages of CRS and HIPEC therapy in the introduction. Their use of HIPEC in a specialist environment is comparatively low. Can they expand on their annual practice referral pattern and decision making concerning management (in broad terms).

2. Can the authors expand a little on their diagnoses of gastric perforations and any delays in diagnosis as this can be a little notorious with a higher mortality when diagnosis is delayed. I thought the tables were excellent and easy to read. I am after a little bit more clinical information in the paper please about their early diagnosis suspicions, basic patient presentations, comments about the ability to make the early diagnosis. Can they comment on the findings at operation ion the perforated cases.

3. They go through the standard analyses of cause in their discussion. For such a paper I think the discussion can be reduced in length by about one quarter. The discussion can be more succinct. I think they could expand on gastroepiploic artery preservation preservation during omentectomy a little more. If they wished to include an image here and had one that would be of more interest.

4. Some of the imagery (although nice) is not necessary and does not add to the quality of the paper.

I enjoyed this well written and well constructed paper. I would favour some minor revisions with a reduction and focus of the discussion, a slight expansion on gastroepiploic arcade preservation and a little bit more clinical information . I would also consider the cogent suggestions of the two reviewers as a minor revision. I would be pleased to see the manuscript revision.

We look forward to receiving your revised manuscript.

Kind regards,

Andrew Zbar

Academic Editor

PLOS ONE

Journal Requirements:

2. In your ethics statement in the manuscript and in the online submission form, please ensure that you have discussed whether all data/samples were fully anonymized before you accessed them and/or whether the IRB or ethics committee waived the requirement for informed consent. If patients provided informed written consent to have data/samples from their medical records used in research, please include this information.

3. In the ethics statement in the manuscript and in the online submission form, please provide additional information about the patient records/samples used in your retrospective study, including: a) the date range (month and year) during which patients' medical records/samples were accessed; b) the date range (month and year) during which patients whose medical records/samples were selected for this study sought treatment.

4. Please consider moving Figures 1 and 2 to the supplementary materials, as the images are quite graphic.

5. Please include your tables as part of your main manuscript and remove the individual files. Please note that supplementary tables (should remain/ be uploaded) as separate "supporting information" files

"Martina Aida Angeles acknowledges the grant support from ”la Caixa” Foundation, Barcelona (Spain), ID 100010434. The fellowship code is LCF/BQ/EU18/11650038."

7. PLOS requires an ORCID iD for the corresponding author in Editorial Manager on papers submitted after December 6th, 2016. Please ensure that you have an ORCID iD and that it is validated in Editorial Manager. To do this, go to ‘Update my Information’ (in the upper left-hand corner of the main menu), and click on the Fetch/Validate link next to the ORCID field. This will take you to the ORCID site and allow you to create a new iD or authenticate a pre-existing iD in Editorial Manager. Please see the following video for instructions on linking an ORCID iD to your Editorial Manager account: https://www.youtube.com/watch?v=_xcclfuvtxQ

Reviewers' comments:

Reviewer's Responses to Questions

**Comments to the Author**

1. Is the manuscript technically sound, and do the data support the conclusions?

Reviewer #1: Yes

Reviewer #2: Yes

2. Has the statistical analysis been performed appropriately and rigorously? 

Reviewer #1: Yes

Reviewer #2: Yes

3. Have the authors made all data underlying the findings in their manuscript fully available?

Reviewer #1: Yes

Reviewer #2: Yes

4. Is the manuscript presented in an intelligible fashion and written in standard English?

Reviewer #1: Yes

Reviewer #2: Yes

5. Review Comments to the Author

Reviewer #1: This is an interesting study looking at possible risk factors for gastric perforation following CRS, mainly in paitents with gynecologic malignancy and peritoneal carcinomatosis. The study is well written and highly detailed and I ongragulate the authors for this work. However, there are still some points in need of clarification prior to publication, in m opinion -

1. The main issue with the manuscript in my opnion is that proton pump inhibitors are not mentioned in the study at all - are they part of the routine treatment following surgical interventions ? were they given to all patients? In addition, I think it is impossible to discuss gastric perforations without mentioning the ongoing debate about the post-operative treatment with PPI.

2. The authors detail the possible reasons why splenectomy is a risk factor although in the univariate analysis distal pancreatectomy was also found to be a significant risk factor. I think that with more patients, pancreatectomy would also be a significant risk factor. Did the authors document pancreatic leaks in thier cohort as well? were there cases in which a pancreatic fistula was the causative factor?

3. I am interested to know whether the authors changed anything in thier practice due to the findings? In addition, i am interested to know if the authors think that in patients with an expected splenectomy, would a selective or completed embolization of the spleen help to avoid such catastrophic complication (to allow collaterel vessels to strengthen the blood supply to the area).

Overeall, this is a well desgined study that I enjoed reading and I would like to thank the authors.

Reviewer #2: The authors present data on the risk of gastric perforation following cytoreductive surgery (CRS). In their cohort of 533 patients, they recorded 13 instances of post operative gastric perforations, amounting to 2.4% of the cohort. The authors correctly point out that there is a paucity of data concerning this rare complication, and indeed they report an incidence higher than previously reported, notably the Sugarbaker paper (ref 17) and the Kyang paper (ref 18).

The main causes found in multivariate analysis were splenectomy and BMI, Although obviously gastric perforation is multifactorial, as for example the combination of splenectomy and infragastric omentectomy would contribute together to gastric devascularization and the threat of ischemia.

I have several comments on the data. First, the patient cohort is heavily skewed towards gynecologic malignancies, (454 of 533, 82%) yet 6 of the 13 patients with gastric perforation have disease of colonic and/or mesothelioma, 46% and not of gyn origin, which I find curious. Although the cytoreductive aspect of surgery should be identical in the differing histologies, one can’t help but notice this aspect of the data and, this should be considered in the discussion.

In addition, although the use of systemic treatment was not found to be statistically significant in perforations, again what jumps out of the data, is that 9/13 or 69% , patients with perforations underwent some form of systemic therapy, either neoadjuvant or HIPEC. It is possible that the small number of perforations from the overall cohort precluded the statistical significance of systemic treatment, nonetheless, it is difficult to categorically exclude systemic therapy from being a major cause of perforation, as this could directly cause poor tissue healing. It is increasingly common for gynecologic malignancies to be treated with a combination of chemotherapy and bevacizumab, which is well known to be a risk factor for poor wound healing and GI perforations, so I would like to know if Avastin was looked at specifically as a possible factor in a multifactorial cascade. This, along with the use of oxaliplatin in the HIPEC cases, could have played some part in tissue disruption, and the small numbers were not powered enough to find statistical significance.

In the discussion, the paragraph on nasogastric suction as a possible cause of perforation is simply rehashing what was found in previous papers, such as the Sugarbaker and Kyang paper, and is sheer conjecture and seems to be dubious. NG drainage has been an integral part of abdominal surgery for many years and has not been directly implicated in gastric perforations, anecdotal case reports notwithstanding, and thus can be omitted.

The photographs in Figure 2 and 3 are superfluous and unnecessary, this paper is directed to a readership of operating surgical oncologists, and they do not need to be shown what a gastric perforation looks like in situ or on CT, kindly omit. Figure 2 looks like almost the same photo shown in the Kyang paper, and there it is to my mind also superfluous. Kindly omit these figures.

Some minor English language editing is needed. Please have a native English speaker edit.

6. PLOS authors have the option to publish the peer review history of their article (what does this mean?). If published, this will include your full peer review and any attached files.

Reviewer #1: **Yes: **Nir Horesh

Reviewer #2: No

---

## [Author Response · Author response to Decision Letter 0]

16 Feb 2021

Response to Reviewers Letter

Dr. Andrew Zbar

Academic Editor

PLOS ONE

Dear Editor,

 Please find enclosed the second version of our manuscript "Risk factors for gastric perforation after cytoreductive surgery in patients with peritoneal carcinomatosis: Splenectomy and increased body mass index", after including the suggested amendments.

You will find below these lines the corrections and observations regarding the revision process. Following your instructions, we are also attaching both a revised manuscript file (clean version) and a tracked changes manuscript file (marked version) illustrating the modifications. Hopefully, they will meet your evaluation criteria.

On behalf of all authors, I want to thank you for considering our paper for publishing. Likewise, we want to show our gratitude to the reviewers for their comments, as they have surely improved our work.

We remain at your complete disposal for any further comment or clarification that you find necessary to address to us. 

Yours faithfully, 

Dr. Gwénaël Ferron

Corresponding author

Comments from the Reviewers:

Reviewer #1:

This is an interesting study looking at possible risk factors for gastric perforation following CRS, mainly in paitents with gynecologic malignancy and peritoneal carcinomatosis. The study is well written and highly detailed and I ongragulate the authors for this work. However, there are still some points in need of clarification prior to publication, in m opinion -

Thank you very much for your revision as it has certainly improved our work.

1. The main issue with the manuscript in my opnion is that proton pump inhibitors are not mentioned in the study at all - are they part of the routine treatment following surgical interventions ? were they given to all patients? In addition, I think it is impossible to discuss gastric perforations without mentioning the ongoing debate about the post-operative treatment with PPI.

We completely agree with the reviewer. It is essential to mention the role of proton-pump inhibitors when evaluating postoperative gastric perforations. The proton-pump inhibitors are part of our routine treatment protocol following cytoreductive surgery. We have added this information to the Methods section (page 7 lines 146-147, underlined the new words):

"Finally, proton-pump inhibitors were systematically administered in the postoperative period."

2. The authors detail the possible reasons why splenectomy is a risk factor although in the univariate analysis distal pancreatectomy was also found to be a significant risk factor. I think that with more patients, pancreatectomy would also be a significant risk factor. Did the authors document pancreatic leaks in thier cohort as well? were there cases in which a pancreatic fistula was the causative factor?

We completely agree with the reviewer in that distal pancreatectomy may have been found to be a significant risk factor for gastric perforation in a larger cohort of patients. However, postoperative gastric perforation is a rare complication and larger cohorts of patients are difficult to obtain.

We have added the following sentences to the Discussion (page 13 lines 269-272, underlined the new words):

"We did not find distal pancreatectomy to be associated with postoperative gastric perforation, which may be explained by the low number of patients in our series undergoing this procedure. Distal pancreatectomy may increase the devascularization of the greater curvature, thereby favoring the risk of gastric perforation."

Among the 13 patients in our cohort who experienced a gastric perforation, none of them had a pancreatic fistula during the postoperative course. Therefore, it is not possible to evaluate its association with gastric perforation in our study.

Moreover, as mentioned in the limitations, we only evaluated the risk factors of one type of complication - gastric perforation - and other complications such as pancreatic fistula were not assessed. It would be interesting to evaluate the risk factors of postoperative pancreatic fistula in future studies.

3. I am interested to know whether the authors changed anything in thier practice due to the findings? In addition, i am interested to know if the authors think that in patients with an expected splenectomy, would a selective or completed embolization of the spleen help to avoid such catastrophic complication (to allow collaterel vessels to strengthen the blood supply to the area).

Yes, we have changed our clinical practice after these findings. Currently, we systematically perform a prophylactic suture of the greater gastric curvature after omentectomy plus splenectomy during cytoreductive surgeries. Even though this measure was not found to prevent postoperative gastric perforation, we believe it is indeed possible and this result was rather due to the low proportion of patients who underwent this prophylactic suture in our series. Moreover, whenever possible, we try to preserve the gastroepiploic arcade during infragastric omentectomy to decrease gastric devascularization.

As stated by the reviewer, it would be interesting to evaluate if preoperative spleen embolization could decrease the risk of gastric perforation by creating collateral vascularization. However, spleen involvement is usually discovered during cytoreductive surgery and we believe it might be difficult to schedule a splenectomy before surgery. Moreover, we have some concerns regarding embolization before the splenectomy as we believe that collateral vascularization may render more difficult the procedure.

Overeall, this is a well desgined study that I enjoed reading and I would like to thank the authors.

We really appreciate all your positive comments which motivate us to keep working and improving our work.

Reviewer #2:

The authors present data on the risk of gastric perforation following cytoreductive surgery (CRS). In their cohort of 533 patients, they recorded 13 instances of post operative gastric perforations, amounting to 2.4% of the cohort. The authors correctly point out that there is a paucity of data concerning this rare complication, and indeed they report an incidence higher than previously reported, notably the Sugarbaker paper (ref 17) and the Kyang paper (ref 18).

The main causes found in multivariate analysis were splenectomy and BMI, Although obviously gastric perforation is multifactorial, as for example the combination of splenectomy and infragastric omentectomy would contribute together to gastric devascularization and the threat of ischemia.

Thank you for your interesting remarks and your comprehensive review of our manuscript.

I have several comments on the data. First, the patient cohort is heavily skewed towards gynecologic malignancies, (454 of 533, 82%) yet 6 of the 13 patients with gastric perforation have disease of colonic and/or mesothelioma, 46% and not of gyn origin, which I find curious. Although the cytoreductive aspect of surgery should be identical in the differing histologies, one can’t help but notice this aspect of the data and, this should be considered in the discussion.

We completely agree with the reviewer regarding this issue. In fact, this association has been evaluated in this study and we found that histology was related to gastric perforation in the univariate analysis. However, in the multivariate analysis, non-ovarian histology did not remain significantly associated with gastric perforation.

As the Editor has requested to shorten the length of the Discussion and to focus on the subject of vascularization, we have not added any comments on the histology to the Discussion.

However, if the Editor or the Reviewer finds it necessary, we can add a few sentences on this issue.

In addition, although the use of systemic treatment was not found to be statistically significant in perforations, again what jumps out of the data, is that 9/13 or 69% , patients with perforations underwent some form of systemic therapy, either neoadjuvant or HIPEC. It is possible that the small number of perforations from the overall cohort precluded the statistical significance of systemic treatment, nonetheless, it is difficult to categorically exclude systemic therapy from being a major cause of perforation, as this could directly cause poor tissue healing. It is increasingly common for gynecologic malignancies to be treated with a combination of chemotherapy and bevacizumab, which is well known to be a risk factor for poor wound healing and GI perforations, so I would like to know if Avastin was looked at specifically as a possible factor in a multifactorial cascade. This, along with the use of oxaliplatin in the HIPEC cases, could have played some part in tissue disruption, and the small numbers were not powered enough to find statistical significance.

We completely agree with the reviewer's comment. We have therefore analyzed if the use of any type of chemotherapy (neoadjuvant chemotherapy [NACT] or HIPEC) was associated with the occurrence of postoperative gastric perforation.

We have included in the univariate analysis the variable systemic therapy (either NACT or HIPEC):

Gastric perforation Yes

n=13 No

n=520 p value

Systemic therapy, n (%) 9 (69.2) 322 (61.9) 0.775

As stated by the reviewer, 69% of women with gastric perforation previously had either NACT or HIPEC. However, among the women without gastric perforation, 62% also had NACT or HIPEC. This difference was not found to be significant.

Among the 13 patients with gastric perforation: 1 had both NACT and HIPEC, 4 had HIPEC, 4 had NACT and 4 did not receive these treatments.

Among the 520 patients without gastric perforation: 50 had both NACT and HIPEC, 44 had HIPEC, 228 had NACT and 198 did not receive these treatments.

We have not added this information to the manuscript as we had already reported the n (%) of patients undergoing neoadjuvant chemotherapy and HIPEC separately. However, if the Reviewer or the Editor finds it necessary, we will gladly add it.

Regarding the use of bevacizumab, it is not biologically plausible for its use to be associated with postoperative gastric perforation, as bevacizumab was not administered before cytoreductive surgery in any patient. Moreover, it is usually administered with adjuvant chemotherapy after the first two cycles. Therefore, none of the patients with a postoperative gastric perforation received bevacizumab before the complication.

In the discussion, the paragraph on nasogastric suction as a possible cause of perforation is simply rehashing what was found in previous papers, such as the Sugarbaker and Kyang paper, and is sheer conjecture and seems to be dubious. NG drainage has been an integral part of abdominal surgery for many years and has not been directly implicated in gastric perforations, anecdotal case reports notwithstanding, and thus can be omitted.

Following the reviewer's suggestion, we have deleted the paragraph on nasogastric suction from the Discussion section (page 15 line 311, deleted words crossed out):

"There are other hypothesis that have been suggested to explain why gastric perforations may occur after these surgeries. After CRS, a nasogastric tube is usually placed, which is relatively rigid and may create a pressure ischemia on the stomach mucosa. As well, the prolonged suction may contribute to the risk of perforation[17,18,30]. Although all the 13 patients with a postoperative gastric perforation had a nasogastric tube without suction placed intraoperatively, we did not have the information of all of the 533 patients included in the study and, therefore, it could not be analyzed as a risk factor."

The photographs in Figure 2 and 3 are superfluous and unnecessary, this paper is directed to a readership of operating surgical oncologists, and they do not need to be shown what a gastric perforation looks like in situ or on CT, kindly omit. Figure 2 looks like almost the same photo shown in the Kyang paper, and there it is to my mind also superfluous. Kindly omit these figures.

We agree with the reviewer that Figures 2 and 3 are unnecessary for oncological surgeons to whom this research is directed. Following the reviewer's request, we have removed Figures 2 and 3, and moved them as Supplementary Figures 2 and 3 (Supporting information).

Some minor English language editing is needed. Please have a native English speaker edit.

Following the reviewer’s recommendation, the English grammar and spelling of our manuscript have been revised by a native English proofreader. All the corrections are shown with the track changes option throughout the manuscript.

Editor:

The authors present from their CRS database of PC the rare complication of gastric perforation with a UVA that suggests a role for BMI, PCI, splenectomy-distal pancreatectomy and primary histology but with BMI and splenectomy remaining of significance on MVA.

I have several caveats:

1. I would expand a little on the outcome advantages of CRS and HIPEC therapy in the introduction. Their use of HIPEC in a specialist environment is comparatively low. Can they expand on their annual practice referral pattern and decision making concerning management (in broad terms).

Due to the large cohort of patients of our study (mainly ovarian cancer patients), we agree with the Editor that it may seem that we have a low rate of HIPEC during CRS.

However, before the publication of the Van Driel et al. study in 2018, HIPEC was not used in our daily clinical practice for ovarian carcinomatosis. Before that study, HIPEC was only employed within clinical trials in ovarian cancer patients.

Since the publication of the OVHIPEC trial, HIPEC is routinely offered to stage III patients undergoing interval cytoreductive surgery after three cycles of neoadjuvant chemotherapy.

As requested by the Editor, we have highlighted the advantages of CRS plus HIPEC, particularly in ovarian cancer, but point out that it was not introduced in clinical practice until the OVHIPEC trial (page 5 lines 88-93, new words underlined and deleted words crossed out):

"Since the recent publication of a randomized trial in stage III ovarian cancer, which showed that the addition of HIPEC to CRS provided a higher recurrence-free and overall survival rate after three cycles of NACT, HIPEC has been introduced to clinical practice[9]. While in ovarian cancer, CRS plus HIPEC is considered an option[9], it represents. HIPEC is also the gold standard for pseudomyxoma peritonei (PMP) and diffuse malignant peritoneal mesothelioma (DMPM)[11,12]."

2. Can the authors expand a little on their diagnoses of gastric perforations and any delays in diagnosis as this can be a little notorious with a higher mortality when diagnosis is delayed. I thought the tables were excellent and easy to read. I am after a little bit more clinical information in the paper please about their early diagnosis suspicions, basic patient presentations, comments about the ability to make the early diagnosis. Can they comment on the findings at operation ion the perforated cases.

As requested by the Editor, we have added some clinical information regarding symptoms presented at diagnosis. In all cases the diagnosis was made using abdominal computed tomography and confirmed during surgery.

We have added this information to the Results section (page 10 lines 205-209, new words underlined and deleted words crossed out):

"The clinical presentation of our patients was a combination of the following signs and symptoms: Acute and severe abdominal pain, abdominal tenderness, nausea, vomiting, gastric fluid in the abdominal drain, fever and/or clinical deterioration. In all cases the diagnosis was made using an abdominal computed tomography (CT) and confirmed during surgery."

The size and location of the perforation found during the surgery appears in the manuscript and is also shown in Table 4 (page 10 lines 209-211):

"The median perforation size was 10 mm (range 2-30) and in all cases the perforation was located at the upper portion of the greater curvature of the stomach (S3 Fig)."

3. They go through the standard analyses of cause in their discussion. For such a paper I think the discussion can be reduced in length by about one quarter. The discussion can be more succinct. I think they could expand on gastroepiploic artery preservation preservation during omentectomy a little more. If they wished to include an image here and had one that would be of more interest.

As suggested by the Editor, we have reduced the length of the Discussion section. All changes can be seen all over the manuscript with the track changes option.

Additionally, we have added the following sentences to the Discussion section in order to lengthen the discussion on gastroepiploic arcade preservation (page 13 lines 280-284, new words underlined):

"As previously mentioned, gastric perforations are commonly located in the upper portion of the greater curvature. Therefore, preserving the gastroepiploic arcade during an omentectomy would reduce the risk of perforation, particularly in patients undergoing this procedure combined with a splenectomy."

Finally, we have added a picture of gastroepiploic arcade preservation during omentectomy (Supplementary Figure 1).

4. Some of the imagery (although nice) is not necessary and does not add to the quality of the paper.

As also requested by the second reviewer, we have removed Figures 2 and 3, and moved them as Supplementary Figures 2 and 3.

I enjoyed this well written and well constructed paper. I would favour some minor revisions with a reduction and focus of the discussion, a slight expansion on gastroepiploic arcade preservation and a little bit more clinical information . I would also consider the cogent suggestions of the two reviewers as a minor revision. I would be pleased to see the manuscript revision.

We really appreciate your encouraging comments and that you are considering our manuscript for publication. As requested, we have lengthened the Discussion on gastroepiploic arcade preservation and added further clinical information:

- "As previously mentioned, gastric perforations are commonly located in the upper portion of the greater curvature. Therefore, preserving the gastroepiploic arcade during an omentectomy would reduce the risk of perforation, particularly in patients undergoing an associated splenectomy." (page 13 lines 280-284, new words underlined)

- "The clinical presentation of our patients was a combination of the following signs and symptoms: Acute and severe abdominal pain, abdominal tenderness, nausea, vomiting, gastric fluid in the abdominal drain, fever and/or clinical deterioration. In all cases the diagnosis was made using an abdominal computed tomography (CT) and confirmed during surgery." (page 10 lines 205-209, new words underlined)

Additionally, we have reduced the overall length of the Discussion and we have made the modifications suggested by the reviewers. All these modifications can be seen all over the manuscript with the track changes option.

Journal Requirements:

We have verified that our manuscript files meet PLOS ONE's style requirements.

2. In your ethics statement in the manuscript and in the online submission form, please ensure that you have discussed whether all data/samples were fully anonymized before you accessed them and/or whether the IRB or ethics committee waived the requirement for informed consent. If patients provided informed written consent to have data/samples from their medical records used in research, please include this information.

We have added the following statement to the Methods section (page 6 lines 118-120, new words underlined):

"All data that could possibly be used to identify individual patients was deleted and all records were anonymized during the retrieval procedure, before the final database was handed to the researchers."

3. In the ethics statement in the manuscript and in the online submission form, please provide additional information about the patient records/samples used in your retrospective study, including: a) the date range (month and year) during which patients' medical records/samples were accessed; b) the date range (month and year) during which patients whose medical records/samples were selected for this study sought treatment.

We have provided the additional information requested by the Journal in the Methods section (page 6 lines 110-116, new words underlined):

A computer-generated search in the institutional patient database was carried out in February 2020 to retrospectively identify all patients who underwent an open upfront or interval CRS after primary diagnosis of PC of different origins (ovarian cancer, endometrial cancer, colon cancer, PMP and DMPM) between March 2007 and December 2018 at the Institut Claudius Regaud Comprehensive Cancer Center - IUCT - Oncopole (Toulouse, France), which is an expert center for rare peritoneal diseases (RENAPE network).

4. Please consider moving Figures 1 and 2 to the supplementary materials, as the images are quite graphic.

As requested by the Journal, Figures 1 and 2 have been added as Supplementary Figures 1 and 2 (Supporting information).

5. Please include your tables as part of your main manuscript and remove the individual files. Please note that supplementary tables (should remain/ be uploaded) as separate "supporting information" files

We have tried to include the tables within the manuscript, but due to Word formatting issues, the tables (mainly Table 4) are not shown as we would like. For this reason, we added a remark indicating where these tables should be placed (e.g., Insert Table 1 here). 

"Martina Aida Angeles acknowledges the grant support from ”la Caixa” Foundation, Barcelona (Spain), ID 100010434. The fellowship code is LCF/BQ/EU18/11650038."

I am sorry for the misunderstanding. This is not funding for this study, but a grant to sustain the fellowship of the first author in the institution where the study took place. However, the grant has been concluded and the main author is exclusively employed by the institution. Therefore, it is no longer necessary to mention the support given by the abovementioned foundation, so we have decided to remove this note from the entire submission.

7. PLOS requires an ORCID iD for the corresponding author in Editorial Manager on papers submitted after December 6th, 2016. Please ensure that you have an ORCID iD and that it is validated in Editorial Manager. To do this, go to ‘Update my Information’ (in the upper left-hand corner of the main menu), and click on the Fetch/Validate link next to the ORCID field. This will take you to the ORCID site and allow you to create a new iD or authenticate a pre-existing iD in Editorial Manager. Please see the following video for instructions on linking an ORCID iD to your Editorial Manager account: 

https://www.youtube.com/watch?v=_xcclfuvtxQ

Following the requirement, we have updated the ORCID iD of the corresponding author in the Editorial Manager.

---

## [Decision Letter · Decision Letter 1]

23 Feb 2021

Risk factors for gastric perforation after cytoreductive surgery in patients with peritoneal carcinomatosis: Splenectomy and increased body mass index

PONE-D-20-33398R1

Dear Dr. Ferron,

We’re pleased to inform you that your manuscript has been judged scientifically suitable for publication and will be formally accepted for publication once it meets all outstanding technical requirements.

Kind regards,

Wen-Chi Chou

Academic Editor

PLOS ONE

Additional Editor Comments (optional):

Reviewers' comments:

Reviewer's Responses to Questions

**Comments to the Author**

1. If the authors have adequately addressed your comments raised in a previous round of review and you feel that this manuscript is now acceptable for publication, you may indicate that here to bypass the “Comments to the Author” section, enter your conflict of interest statement in the “Confidential to Editor” section, and submit your "Accept" recommendation.

Reviewer #1: All comments have been addressed

Reviewer #2: All comments have been addressed

2. Is the manuscript technically sound, and do the data support the conclusions?

Reviewer #1: Yes

Reviewer #2: Yes

3. Has the statistical analysis been performed appropriately and rigorously? 

Reviewer #1: Yes

Reviewer #2: Yes

4. Have the authors made all data underlying the findings in their manuscript fully available?

Reviewer #1: Yes

Reviewer #2: Yes

5. Is the manuscript presented in an intelligible fashion and written in standard English?

Reviewer #1: Yes

Reviewer #2: Yes

6. Review Comments to the Author

Reviewer #1: (No Response)

Reviewer #2: After going over the revised manuscript, I think the authors have addressed the concerns of the reviewers and editor. And I can say that the manuscript is much better now in its current form and advise to accept.

7. PLOS authors have the option to publish the peer review history of their article (what does this mean?). If published, this will include your full peer review and any attached files.

Reviewer #1: **Yes: **Nir Horesh

Reviewer #2: No

---

## [Editor Report · Acceptance letter]

24 Feb 2021

PONE-D-20-33398R1 

Risk factors for gastric perforation after cytoreductive surgery in patients with peritoneal carcinomatosis: Splenectomy and increased body mass index  

Dear Dr. Ferron:

I'm pleased to inform you that your manuscript has been deemed suitable for publication in PLOS ONE. Congratulations! Your manuscript is now with our production department. 

Kind regards, 

on behalf of

Dr. Wen-Chi Chou 

Academic Editor

PLOS ONE